# Mechanisms of the *FMR1* Repeat Instability: How Does the CGG Sequence Expand?

**DOI:** 10.3390/ijms23105425

**Published:** 2022-05-12

**Authors:** Elisabetta Tabolacci, Veronica Nobile, Cecilia Pucci, Pietro Chiurazzi

**Affiliations:** 1Dipartimento Scienze della Vita e Sanità Pubblica, Sezione di Medicina Genomica, Università Cattolica del Sacro Cuore, Fondazione Policlinico Universitario “A. Gemelli” IRCCS, 00168 Rome, Italy; elisabetta.tabolacci@unicatt.it (E.T.); veronicanobile88@gmail.com (V.N.); c.pucci91@yahoo.it (C.P.); 2UOC Genetica Medica, Fondazione Policlinico Universitario “A. Gemelli” IRCCS, 00168 Rome, Italy

**Keywords:** *FMR1* gene, CGG repeat, mechanisms of instability, dynamic mutations, repeat expansion disorders, molecular medicine, neurological disease

## Abstract

A dynamic mutation in exon 1 of the *FMR1* gene causes Fragile X-related Disorders (FXDs), due to the expansion of an unstable CGG repeat sequence. Based on the CGG sequence size, two types of *FMR1* alleles are possible: “premutation” (PM, with 56-200 CGGs) and “full mutation” (FM, with >200 triplets). Premutated females are at risk of transmitting a FM allele that, when methylated, epigenetically silences *FMR1* and causes Fragile X syndrome (FXS), a very common form of inherited intellectual disability (ID). Expansions events of the CGG sequence are predominant over contractions and are responsible for meiotic and mitotic instability. The CGG repeat usually includes one or more AGG interspersed triplets that influence allele stability and the risk of transmitting FM to children through maternal meiosis. A unique mechanism responsible for repeat instability has not been identified, but several processes are under investigations using cellular and animal models. The formation of unusual secondary DNA structures at the expanded repeats are likely to occur and contribute to the CGG expansion. This review will focus on the current knowledge about CGG repeat instability addressing the CGG sequence expands.

## 1. Introduction

Instability of short repeated sequences underlies the so-called dynamic mutations responsible for more than 40 mostly neurological clinical conditions also known as repeat expansion disorders or REDs [1,2]. REDs are intrinsically dynamic, i.e., repeated sequences (tri-, tetra-, penta-nucleotides and so forth) which continue to mutate across generations and within tissues of an individual, while “normal” mutations (e.g., single nucleotide variants or copy number variants) are transmitted to offspring and are retained in somatic tissues without further changes. REDs also often display genetic anticipation, a phenomenon by which disease severity increases and age of onset decreases from one generation to the other, in parallel with expansion of the repeat sequence. Most REDs are caused by a trinucleotide repeat expansion, as happens in Fragile X syndrome (FXS; OMIM #300624), the first such condition, identified in 1991. FXS is a very frequent cause of monogenic intellectual disability (ID), due to the expansion of a CGG tract in exon 1 of the *FMR1* gene, located in Xq27.3. When expansion exceeds 200 repeats (FM, full mutation), cytosines in the CGG repeat and in the neighboring CpG island become methylated (MFM, methylated FM) and the *FMR1* gene is silenced [3]. FXS is eventually caused by a loss-of-function effect produced by the dynamic mutation, causing local epigenetic modifications and absence of the *FMR1* protein, FMRP. The estimated frequency of FM in the general population is 1:7143 males (who will be affected by FXS) and 1:11,111 females, who present a variable phenotype ranging from learning disability or mild ID to a full-blown phenotype like that of affected males [4,5]. To perform a high sensitivity diagnostic examination for FXS, the gold standard should include CGG sizing and methylation analysis, the latter is particularly indicated during prenatal diagnosis [6].

The CGG repeat is highly polymorphic in length, with a mode of 30 triplets in the general population, and it is usually interrupted by an AGG every 9-10 CGG triplets [7,8]. AGG interruptions have no effect on *FMR1* transcription efficiency [9]. Based on size, the CGG sequence may be classified in four allele categories, each with a different degree of instability through generations and within tissues. Normal alleles (5–44 repeats) are stably transmitted from parents to children, although rarely very small repeat size changes are detected particularly during paternal transmission. Intermediate or gray zone alleles (45–55 repeats) sporadically undergo length instability in some families, again with a greater instability in paternal than maternal transmissions [10,11]. The greatest instability is well documented for premutation alleles (PM, 56–200 repeats), that expand to full mutation (FM, >200 repeats) during maternal meiosis, causing FXS to manifest in the offspring. Until now, the smallest PM known to have expanded to FM in a single maternal meiosis had 56 repeats [12]. The risk of passing FM to the offspring increases with the length of maternal PM and it is nearly 100% for mothers with >90 CGGs [13]; it also seems that increasing maternal age directly correlates with the increase in PM-to-FM expansion risk [14].

This increase in risk of having more FXS children in parallel with the increase in length of the CGG repeat as generations pass, is known as the Sherman paradox [15]. Hence, the number of FXS affected individuals tends to increase over generations. An actual genetic anticipation phenomenon, i.e., a younger age of onset with passing generations, as described in myotonic dystrophy, is not described in FXS families. The so-called “all or nothing” effect consists of having the classical FXS phenotype when an MFM is transmitted, no matter how large it is, since the phenotype depends on the epigenetic changes occurring when the CGG expansion exceeds 200 units.

Paternal PM alleles are typically transmitted to all their daughters without expanding to FM, although extremely rare cases of father to daughter FM transmission have been reported [16,17]. Furthermore, although FXS males typically do not have offspring, it is expected that all their daughters would inherit a PM not a FM, since surprisingly it appears that FM contracts to PM in the germline of affected males [18]. On the contrary, females with a FM have 50% risk of transmitting it to their offspring, whether male or female, possibly with a larger CGG expansion.

It is important to point out that PM alleles are not only more meiotically unstable but also associated with a late-onset neurodegenerative condition known as Fragile X Tremor/Ataxia Syndrome (FXTAS, OMIM #300623), that affect 33% of carrier males and approximately 5–10% of carrier females [19,20]. Premutated females are also at risk of developing a premature menopause, called Fragile X-associated primary ovarian insufficiency (FXPOI, OMIM # 311360) [21]. The estimated prevalence in the general population of males carrying a PM allele is 1:855, while that of carrier females is 1:291 [5]. It is also worth of note that FXTAS and FXPOI are never observed in FXS patients, since the underlying mechanism is different: FM alleles are silenced and both *FMR1* mRNA and FMRP are lacking (loss-of-function), while PM alleles are transcribed and the expanded CGG repeat in the 5′ untranslated region of the mRNA causes neurodegeneration with at least two gain-of-function mechanisms, namely, RNA toxicity and production of abnormal polypeptides due to repeat-associated non-AUG (RAN)-dependent translation [22].

The presence of AGG interruptions inversely correlates with the risk of transmitting a FM to the offspring if a mother carries a PM: more interspersed AGGs in a PM allele correlate with a lower risk of FM transmission, i.e., of PM-to-FM expansion in maternal meiosis. For example, a PM allele with 75 repeats with no interspersed AGGs has a risk of expansion to FM of 77%, but the risk is lowered to 12% for PM of the same length with two AGG interruptions [23]. Interestingly, loss of AGG interruptions in the CGG repeat seems to occur after contraction events of a maternal PM [11].

In fact, although the instability of the *FMR1* repeat mostly consists of expansions of the CGG sequence, contractions to smaller length are observed in all classes of alleles [11]. Expansion and contraction events both contribute to the repeat size mosaicism frequently displayed by PM and FM alleles [24]. Contractions occur more frequently in paternal than maternal transmissions, and the frequency of paternal contractions increases linearly with repeat size, being more frequent in PM range. Most paternal PMs contract to a slightly shorter PM, but maternal PM alleles may also be transmitted as PM, intermediate, or even normal alleles [25,26].

Finally, it is important to remember that mitotic instability of FM alleles very often results in somatic mosaicism with cells of the same individual carrying CGG repeats of different size and sometimes even different methylation status [27,28]; for example, a postmortem study of a high-functioning male demonstrated the presence of an unmethylated mutation with a wide span of sizes (from PM to FM) in leukocytes and most brain areas, while in the cortex of the parietal lobe and in other non-brain tissues a methylated FM of a single size was found [29].

In summary, from the unbiased observation of FXS families, we conclude that three main factors influence the stability of the *FMR1* repeat: its size, its internal structure (AGG interruptions) and the sex of the transmitting parent. Smaller PM alleles with two (or more) AGGs are more stable than longer alleles, particularly if the latter are transmitted through oogenesis. Several hypotheses have been made to explain the apparently exclusive maternal transmission of FXS starting from a PM (Figure 1): is a maternal meiotic event occurring during gametogenesis? Or does an early embryonic mitotic event cause expansion only of a maternal PM? Do both meiotic and mitotic events contribute to the maternal PM-to-FM expansion? Which (different) mechanisms are responsible for the phenomenon of instability in the germline and in somatic cells? To answer these questions, we discuss the relevant literature, published so far, describing how errors during DNA replication, repair or recombination lead to the instability of the CGG repeat. Errors in these processes are largely caused by secondary structures formed on DNA strands with expanded repeats.

## 2. The FRAXA Fragile Site and Secondary Structures at the CGG Repeat

In 1969, Lubs wrote, “The marker X chromosome described in this paper offers an additional mechanism by which the expression of the effects of a chromosomal variant may vary from generation to generation or individual to individual and permits several basic observations about the behavior and identification of the human X chromosome” [30]. He was the first to describe a marker folate-sensitive fragile site (later denominated FRAXA) on the long arm of chromosome X (Xq27.3) in males affected by ID and their unaffected mothers of a three-generations family. Only in 1991 the *FMR1* gene with its polymorphic CGG tract was cloned in this position [3]. FRAXA belongs to the rare fragile sites, whose main feature is the late replication during S phase of the cell cycle for normal size alleles, but in the G2/M phase for FM alleles. Under folate deprivation, which further delays replication, cells enter mitosis without having completed replication of the *FMR1* region and display the FRAXA fragile site [31,32,33]. Delayed replication of FM alleles has been linked to the expanded CGG repeat, which may form secondary structures (“hairpins”) that inhibit/retard progression of replication fork and may facilitate “replication slippage”, allowing further expansion of the repeat itself [34]. Such secondary structures may be formed by virtually all repeated sequences and may be at the heart of the instability mechanisms of all REDs [35]. Actually, the expanded CGG tract of the *FMR1* gene may form a number of secondary structures, such as hairpins, triplex and quadruplex. Single-stranded DNA composed of repeated CGGs may form hairpin structures involving both Watson–Crick base pairs and mismatched base pairs. Triplex structures called R-loops may also form during transcription of long GC-rich repeats, when nascent RNA anneals with template DNA producing a Watson–Crick RNA:DNA hybrid and the non-hybridized DNA strand displaces as a single strand (ssDNA) [36]. R-loops, and thus active transcription, contribute to instability because the unbound strand in turn may itself form secondary structures. Moreover, single-stranded CGG repeats can fold into quadruplex structures stabilized by G quartets, where four guanines are linked by Hoogsteen hydrogen bonds forming thermodynamically stable four-stranded structures. Conversely, cytosines in the CCG strand can form i-motifs, another four-stranded structure stable in acidic environment. Thus, the *FMR1* coding strand containing CGG repeats forms G-quadruplex, while the non-coding strand with its CCG repeat will rather form an i-motif. Recent studies demonstrate formation of quadruplex structures in the CGG DNA (sense strand of the *FMR1* gene) and its transcript, while the corresponding antisense DNA strand (CCG) and the antisense transcript prefer hairpin structure and/or an i-motif conformation, respectively (Figure 2) [37]. In addition to these atypical structures, it should be remembered that the CGG•CCG duplex can adopt a left-handed Z-DNA conformation [38]. Not surprisingly, it has been shown that AGG interruptions modulate the structure of the expanded CGG repeat [39], diminishing the formation and stability of tetrahelical structures [40].

Hairpins and quadruplex structures formed by the repeated CGG tract during replication and/or transcription are thought to contribute to the instability of the sequence and ultimately to the expansion of the repeats; in turn, further expansion generates more secondary structures and more instability. Destabilizing such structures with specific compounds may contribute to mitigate the block of DNA replication and, possibly, could support the reactivation of *FMR1* gene transcription in FXS cells.

Human DNA helicases of the RecQ family, such as the Werner syndrome helicase/exonuclease (WRN), and heterogeneous nuclear ribonucleoproteins (hnRNPs), mainly CArG-binding factor A (CBF-A), were specifically demonstrated to unwind and destabilize quadruplex structures of CGG repeats [41,42,43]. It should be remembered that chromosomal fragility, i.e., fragile site FRAXA expression, is observed under replicative stress conditions (e.g., folate deprivation), when molecular mechanisms helping the replication fork advance along the expanded CGG repeat fail. Indeed, human DNA helicase B (HDHB) is enriched in S-phase and found localized at replication forks stalled in repetitive sequences of common fragile sites as well as the *FMR1* CGG repeat under non-stressful conditions, apparently preventing replication forks stalling [44]. How the “lesions” at fragile sites are repaired is poorly understood. Evidence suggests a role for homologous repair (HR) [45], although no protein involved in HR has been directly identified in the maintenance of fragile site stability. Delayed FRAXA locus replication under folate deprivation leads to failure of sister chromatids segregation at anaphase, generating further instability on entire X chromosome [33]. A higher sensitivity to chromosome breakage along the X chromosome was supported by the accumulation of micronuclei (biomarkers of DNA damage) in FXS patients compared with controls [46]. Delayed replication should also lead to activation of cell-cycle checkpoints related to DNA damage response, namely Ataxia-Telangiectasia and Rad3-related (ATR) and Ataxia-Telangiectasia Mutated (ATM) proteins that protect the integrity of replicating chromosomes and respond primarily to DNA double-strand breaks (DSBs), respectively. ATR-deficient cells were very sensitive to aphidicolin (a DNA polymerase inhibitor) and showed a very significant increase in gaps and breaks at fragile sites [47]. Conversely, ATM-deficient cells did not display increased fragile site expression, suggesting that DSBs are not their primary cause. Further evidence suggests that aphidicolin does not induce FRAXA and other fragile sites containing CGG repeats, but treatments that negatively impact thymidylate synthase, and thus the size and composition of nucleotide pools available for replication, may do. Fluorodeoxyuridine (FdU, an inhibitor of thymidylate synthase) increases the presence of FRAXA sites to such extent that prolonged treatments result in high frequency loss of the entire X chromosome carrying the FM allele, linking replication problems, fragile site expression, and aneuploidy [33]. FdU increases both the number of γ-H2AX foci (a marker of DNA damage) in normal and FXS patient cells and the frequency of *FMR1* gene co-localization with these foci in FXS cells. Combining FdU with KU55933, an ATM inhibitor, a reduction in chromosome fragility is noticed, suggesting that ATM contributes to FdU-induced chromosome fragility. Overall, these data support the hypothesis that FRAXA displays an alternative form of chromosome fragility in absence of FdU, which is normally prevented by an ATM-dependent process [48]. ATM protein and intracellular calcium signaling have been found increased in brain of mouse model of PM, suggesting a greater vulnerability to apoptotic activation and a possible link with neurodegeneration observed in FXTAS [49].

Chakraborty et al. [50] reported an increased accumulation of DSBs in FXS cell lines compared to unaffected control and DSBs appeared to co-localize with R-loop forming sequences. The exogenous re-expression of FMRP in FXS fibroblasts reduces R-loop-induced DSBs, suggesting that FMRP also plays a role in genome maintenance, preventing accumulation of R-loops.

In FXS fibroblasts and lymphoblastoid cells under folate stress, the FRAXA locus undergoes to DNA synthesis in mitosis (MiDAS), a process occurring through break-induced DNA replication and that requires the SLX1/SLX4 endonuclease complex, RAD51 recombinase and POLD3, a subunit of polymerase delta [51]. These authors reported failure to complete MiDAS at FRAXA site leading to severe locus instability and anomalous segregation during mitosis. Inhibition of MiDAS prevents chromosome fragility but increases the frequency of abnormal chromosome segregation. The fragility is the result of the delayed chromosome condensation that occurs when MiDAS fails to complete in time (Figure 2).

The interplay between instability, fragile site formation and late replication is more apparent in cells with high rate of cell division and faster replication, but less obvious for female gametes and neurons.

## 3. Mouse Models of CGG Instability

In the last two decades, a large body of literature has been published exploring factors and mechanisms which affect repeat (in)stability, although it is difficult to model some events in animal models. Transgenic mice have been developed to study the repeat expansion mechanism and FXS pathology caused by CGG expansions, and, in particular, *Fmr1* knock-in (KI) mice have been intensively studied [52,53,54,55,56,57,58]. The *Fmr1* KI mice may recapitulate some features of repeat instability since humans and mice share some molecular basis. In fact, it seems that both species show expansions and contractions events, even if expansions are more frequent than contractions in the PM range. Furthermore, expansion only occurs when the PM allele is on the active X chromosome; thus, expansion is found in approximately 50% of cells in females with normal X chromosome inactivation, confirming that DNA methylation has a protective effect on instability [57]. In addition, in mice and humans, expansions occur during male and female gametogenesis, even if expansions in male mice are observed in the spermatogonia stem cells, cells with high proliferation rate, while in female mice expansions occur in non-dividing oocytes [59]. As in humans, maternal age has also been reported to affect PM expansion in *Fmr1* KI mice, since more expansions were found in older oocytes than younger ones. However, since human reproductive lifespan is longer than that of mice, human oocytes may accumulate more CGG repeats, resulting in very large FM alleles [59]. Moreover, *Fmr1* KI mouse models are subject to other genetic modifiers that affect repeat instability in other human REDs [60,61]. The expansion mechanisms probably involve DNA damage repair or recombination events rather than chromosomal replication failure, suggesting cell proliferation is not exclusively required for expansion. In fact, it seems that different cell types have different propensity for expansion in *Fmr1* KI mice. Organs, such as heart, show very little expansion compared to testes and liver, where expansions continue to accumulate [58,59]. The high level of expansion in the brain of *Fmr1* KI mice compared to little expansion in blood suggests that in some FXS patients, the repeat size expansion found in blood may not correspond to what is found in cells related to the pathology, i.e., neurons [58]. Several proteins involved in DNA repair pathways have been identified to promote or protect against expansions in *Fmr1* KI models. For example, mutations in mismatch repair (MMR) proteins, including MSH2, MSH3, MSH6 and MLH3, affect the extent of expansion by reducing expansions altogether [57,62,63,64]. MSH3 (the MSH2-binding partner in the MutSβ complex) is required for 98% of germ line expansions and all somatic expansions in mice increase the stability of CCG-hairpins. Two different contraction mechanisms operate in the mouse model, a MutSβ-independent one that generates small contractions and a MutSβ-dependent one that generates larger ones [65]. It seems that two exonucleases, EXO1 and FAN1, play a protective role: if they are mutated, the expansion rate increases. EXO1 affects expansions in the germ line but not in the brain [63], contrary to FAN1 [66]. Recently, it was found that the expansion mechanism competes with non-homologous end-joining (NHEJ) repair for the processing of a DSBs intermediate, while a protective role against expansion is played by LIG4, a ligase essential for NHEJ [67]. Finally, in this mouse model, the same genetic factors required for expansion in dividing cells are required for expansion in non-dividing ones.

## 4. CGG Instability during Gametogenesis and in Tissues with Low Rate of Cell Division

However, not just how, but also when and where PM expansion exactly occurs it is still being debated. It has been proposed that PM-to-FM expansion does not occur during meiosis but in an early post-zygotic stage [68,69]. Two main models emerge from scientific literature that explain PM-to-FM expansion: a prezygotic model, in which PM expansion would occur during recombination events in maternal germline and later expands during early embryo development; a postzygotic model, in which an oocyte carrying a PM would expand to FM during the very first stages of embryogenesis before primordial germ cell (PGC) segregation (Figure 1) [70]. Females carriers of a PM allele, in fact, show the presence of the expansion already in pre-implantation embryos [71,72]. Not even studies on embryonic stem cells (ESCs) isolated by FXS fetuses that preserve FM expansion are diriment to solve this question [73,74,75]. Assuming the postzygotic model, an embryo carrying a FM derived from maternal PM should exhibit PM expansions in gametes and not in other tissues, since the sex of embryo is not determined before PGC segregation. From the analysis of intact ovaries and testes derived from FM fetuses of PM mothers, CGG expansions in the range of PM were not detected in gametes, but only FM were found as in other somatic fetal tissues (i.e., brain, skin and lung) [76]. Only a 17-week testis showed a faint PM signal with very low levels of FMRP expression. Together these data disproved the mitotic model of CGG expansion in the early embryonic stages sparing the germline. The PM detected in the sperm of FXS affected males may derive from a positive selection mediated by mitotic contraction events, taking place in spermatogonia. Furthermore, in the paper by Malter and co-authors [76] a relevant issue has been addressed: the FM expansion precedes its methylation in both males and females. In fact, the authors detected an unmethylated FM in fetal oocytes and concluded that the PM-to-FM expansion event may occur in the maternal germline or very early in embryogenesis, prior to *de novo* DNA methylation. Since instability of the repeated tract involves both maternally derived expansions and contractions in gametes of adult FM males, it has been proposed that the absence of aberrant CpG methylation, such as in PGCs, may enhance repeat deletions through an undefined process. Two articles of the same group addressed this matter in 2015, using an SV40 primate replication system containing a CGG expansion and bidirectional ORIs (origin of replication sites), which were methylated using *Sss*I methylase, when necessary [77,78]. With this model they demonstrated that CGG contractions occur during replication and depend on the length of the CGG repeat and CpG methylation. Precisely, the proximity of ORI to the CGG tract and the positioning of the repeat on the lagging strand turned out to be the principal determinants of CGG contractions, while the role of DNA methylation is to stabilize the repeated sequence. Taken together, these results make parental gender the major risk factor for the transmission of FXS, although not the only one.

Since expansion seems to occur in non-dividing cells such as oocytes and neurons, DNA repair systems may be involved in the pathogenic mechanism. One model suggests a role for base excision repair (BER) of 7,8-dihydro-8-oxoguanine (8-oxoG), the most common oxidation product of DNA (Figure 3A). During repair events, strand slippage or displacement may generate hairpins that are bound by the MutS proteins [79]. Formation of secondary structures such as hairpins may trigger multiple rounds of BER, since guanines included in the hairpins are susceptible to DNA damage and are difficult to repair. The role of BER in repeat expansions is supported by the observation that mutations in *OGG1* and *NEIL1*, encoding DNA glycosylases involved in 8-oxoG repair, reduce expansions in a mouse model of Huntington disease [80,81]. The BER mechanism may also play a role during transcription, when single-stranded DNA regions form R-loops that are more susceptible to oxidative damage, generating 8-oxoG. Expansion of the *FMR1* gene located on the active X chromosome may be due to the BER mechanism [57].

On the other hand, contraction events seem to follow different pathways (Figure 3B). Contractions of the CGG repeat are apparently very common during spermatogenesis, which may explain why PM males do not transmit a FM to their daughters and why FXS males have only PM alleles in their sperm. Methylated FM alleles do contract [82], thus some contractions are transcription independent, but small contractions seem to be related to DNA synthesis. It has been proposed that hairpin formation on the repeated template strand cause strand-slippage during replication, thus the nascent strand includes fewer repeats than the template one. A second round of DNA replication would generate a small, contracted allele. Conversely, larger deletions characterized in normal, PM and FM carriers present with breakpoints associated with microhomologies (MHs) of 2–9 nucleotides [83,84], which may be responsible of a process known as microhomology mediated end joining (MMEJ). MMEJ represents a repair mechanism of a DSB and starts from an end-resection that reveals MHs at both sides of the break. Non-homologous flaps will be removed allowing the annealing of MHs strands, while gaps will be filled by Polθ or Polβ and the ends will be ligated by either ligase 3 (Lig3) or ligase 1 (Lig1) [85].

Mechanisms other than repair, such as those related to DNA synthesis and active transcription (see below), cannot be completely excluded also in cells with low rate of division.

## 5. CGG Instability in Tissues with High Rate of Cell Division

Although CGG expansions display their highest instability in the germline, also the analysis of cultured single lymphocytes exhibited modest mitotic instability, confirming that in FXS the CGG repeat is unstable both through germline transmission and in somatic cells during the lifetime of the individual [11,86]. In other words, some post-zygotic mitotic instability occurs, and specific tissues show heterogeneous degrees of CGG size mosaicism [87]. The length of the expanded repeat remains the main determinant of FXS, but both in cis elements (localized in the expansion itself) and in trans elements (physically distant from the CGG repeat) may contribute to instability, as summarized in Figure 4.

The first in cis elements are the length of the CGG repeats and the presence of interspersed AGGs. The ability of the unstable DNA to expand in somatic cells is again likely due to the secondary structures formed by the repeated DNA: if the CGG sequence is longer (possibly without interrupting AGGs), more hairpins will form on the nascent strand and the repeat may increase more in size. The main expansion model is associated with replication fork stalling and restarting with polymerase slippage (Figure 4). The presence of unwound DNA during replication allows secondary structures formation (most likely guanine quadruplexes), which, if not correctly recognized by the repair mechanisms, would favor contractions or expansions, depending on which strand (template or nascent) they will be formed [88]. Slipped-strand DNA mispairing may be favored also during transcription through the same mechanisms [89]. During DNA replication, the probability of expansions or contractions are related with the CGG repeat orientation relative to the ORIs. In the normal CGG tract of the *FMR1* gene, two ORIs on both strands are active, hence expansions and contractions happen with the same probability and repeat size remains stable. If only one ORI works, as it has been reported in FM alleles [90,91], replication leads to secondary structures formation on the nascent strand containing CGG repeats and consequent expansion of the tract. According to the “origin switch” model, the location of the closest ORI to the repeat changes with a consequent switch in the replication fork direction. Consequently, secondary structures will form in the lagging strand and not in the leading one, promoting CGG instability. In the “fork shift” model, instead, altered replication fork progression might shift the location of the Okazaki initiation zone relative to the repeat tract, allowing the formation of a mutagenic secondary DNA structure and ultimately leading to repeat instability [1,89]. DNA slippage and secondary structure formation on the newly synthesized strand, followed by reannealing on the repeated sequence, will generate an expansion in the repeat tract that will not be compensated by contraction events [1,35,89,92,93]. Leading- and lagging-strand behave differently during replication; a possible explanation could be that Polε and Polδ respond differently to replication stress [94]. As McMurray reviewed in 1995, DNA stands at a crossroad in presence of secondary structures: maintaining or repairing the heteroduplex molecule. Failure in repairing the secondary structure results in non-Mendelian segregation of the repeat sequence at the first mitotic division. Consequently, the ratio of the insertion allele to the original allele will be higher in the daughter cells during mitosis [93]. Hairpin and G-quadruplex must be considered as barriers during replication, and triggers of genome instability, although the precise mechanism through which the CGG repeat causes fork stalling remains yet unclear.

Another cis-acting factor shown to influence repeat instability is transcription: bidirectional transcription, known to occur at the *FMR1* locus [95], increases instability compared to unidirectional transcription [96]. The transcription complex, passing through the expansion, leave the non-template strand uncoupled, promoting the hairpin and quadruplex formation and consequently increasing instability [24].

Chromatin structure is a further cis-element: histone modifications, DNA methylation, chromatin binding proteins and chromatin looping may all affect repeat instability by influencing transcription levels, facilitating the formation of secondary structures, or promoting access to DNA by repair/recombination machinery [24,97].

To summarize, several cis-modifiers contribute to increasing the instability of the CGG expanded repeat, such as AGG interruptions, secondary structure formation, ORI position, transcriptional activity (both sense and antisense) and chromatin organization.

To complicate the picture, it should be considered that secondary structures are closely related to trans-modifiers and that DNA replication and repair mechanisms can play together to increase instability [98]. Recently, it was observed that proteins involved in DNA repair, such as MSH2, MLH3, MSH6, PCNA, RPA1, FEN1, LIG1, HMGB1, Polβ, influence trinucleotide repeat instability [24]. For example, Polβ, a protein involved in BER, was proved to be involved in CGG repeat expansion in the FXS murine model [99]. BER could facilitate CGG instability by repairing the DNA oxidative damage. Furthermore, differential expression of transcripts encoding components of DNA repair and DNA replication has been detected in FXS patients relative to control individuals [100].

Recently, Kononenko et al. [101] proposed a role for break-induced replication (BIR) in the expansion of the *FMR1* CGG repeat: replication forks stalling cause SMARCAL1 recruitment to promote forks resolution; after isomerization and enzymatic resolution of the Holliday junction, fork regression is converted into an one-ended DSB, with only one free end that will be repaired by break-induced replication (BIR). This single strand can invade the sister chromatid with the help of RAD52 and RAD51 and the newly formed loop is extended by Polδ; given that the invasion of the repetitive strand is out of register, it can easily result in expansion, contraction or mutagenesis [101]. Furthermore, it has been proposed that unstable DNA repeats dramatically increase mutation rates in surrounding DNA segments and that these mutations can occur up to kilobases away from the repetitive tract itself: such a process has been called RIM, i.e., repeat-induced mutagenesis [102].

Finally, three other mechanisms should be mentioned that may promote repeat expansions: failure to remove displaced 5′ DNA flaps and their subsequent incorporation into DNA, misalignment during recombinational repair and synthesis during DSB repair.

## 6. Concluding Remarks

Despite the amount of experimental data produced so far, indicating that the determinants of *FMR1* gene instability reside mainly in the CGG repeat itself, more research is needed to better understand all the mechanisms responsible for expansion and contraction events and their timing. In fact, only precise knowledge of the pathogenetic pathways will potentially suggest tailored therapeutic intervention. For example, recent works targeted small compounds to CGG secondary structures in order to interfere with their deleterious effects; indeed, antisense oligonucleotides and small molecules targeting CGG-RNA hairpins have been shown to reduce R-loop formation with beneficial effects on the *Fmr1* KI mouse models [103,104,105,106]. Although an in vivo reduction in FM alleles in FXS patients seems unrealistic, detailed understanding of the *FMR1* CGG repeat instability and consequent gene silencing may disclose new therapeutic approaches to treat Fragile X-related disorders.

## Figures and Tables

**Figure 1 ijms-23-05425-f001:**
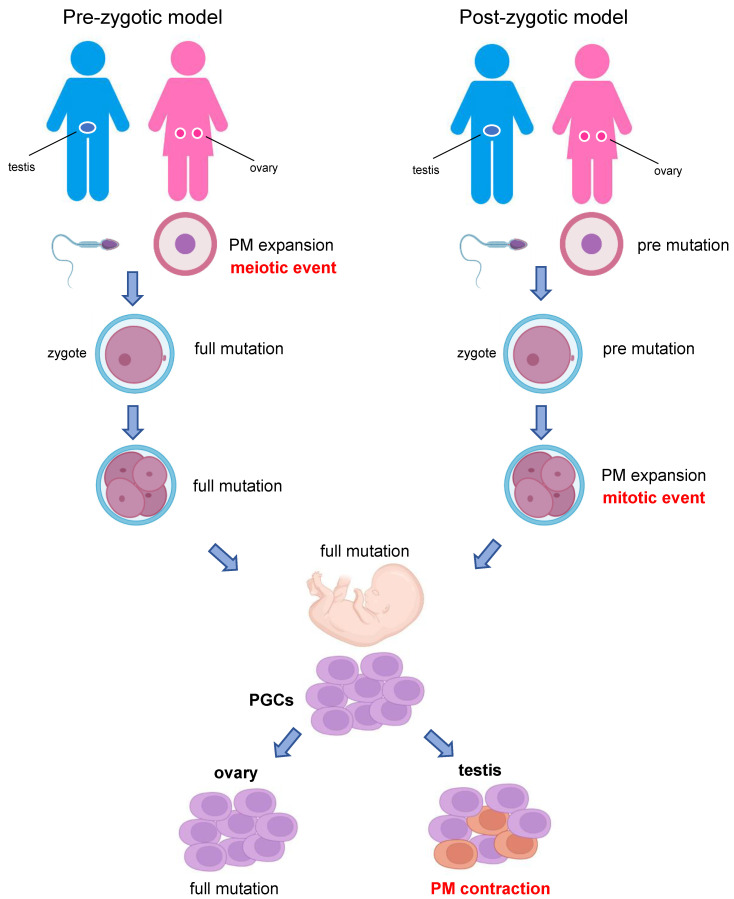
Two models to explain PM-to-FM expansion. The prezygotic model speculates that PM would jump to FM only during maternal germline (meiosis), possibly during recombination, and would further expand during early embryo development (on the left). The postzygotic model hypothesizes that an oocyte carrying a PM would expand to FM during the very first stages of embryogenesis (mitosis) before primordial germ cell (PGC) segregation (on the right). This model does not readily explain why a PM allele transmitted by a father to his daughters cannot expand to FM.

**Figure 2 ijms-23-05425-f002:**
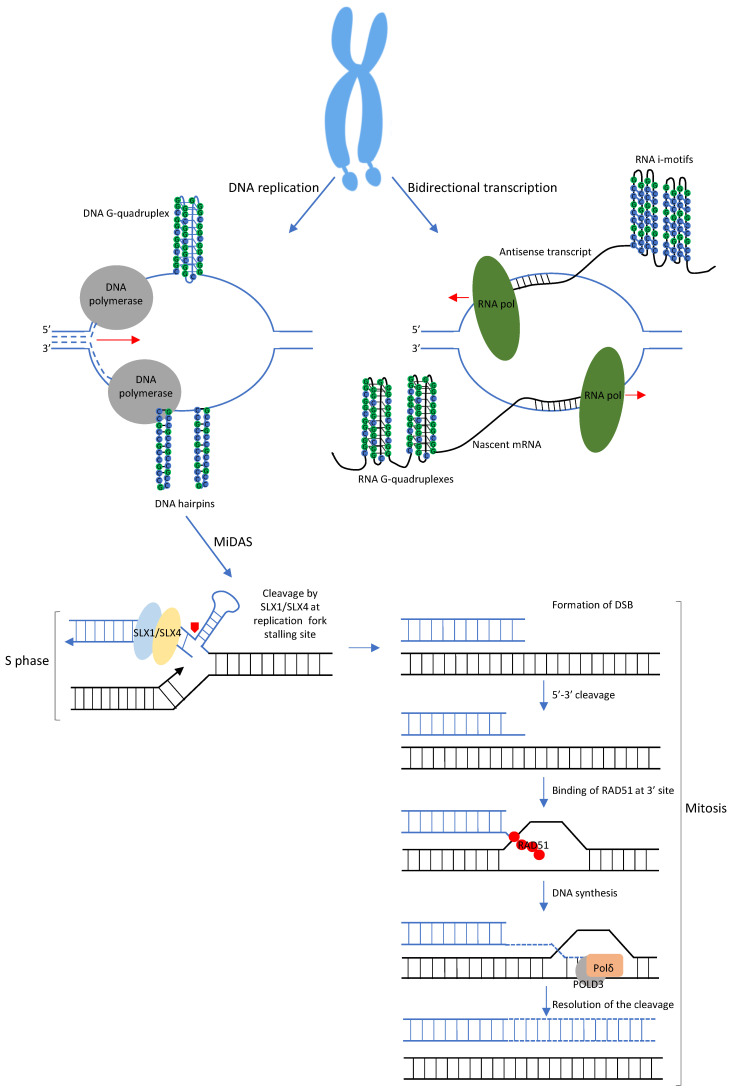
Secondary structures forming at the FRAXA site and their role during DNA synthesis in mitosis (MiDAS). During DNA replication at the FRAXA site, two different secondary structures may form in the CGG repeat sequence: G quadruplex structures on the leading strand and hairpins on the lagging strand. During DNA bidirectional transcription, both nascent RNAs may assume anomalous conformations: the *FMR1*-mRNA, that is G-rich, is able to form G-quadruplexes, contrariwise the antisense transcript, C-rich, prefers i-motifs. The formation of several secondary structures at the expanded CGG sequence cause failure to complete MiDAS at FRAXA locus. MiDAS starts during S phase of cell cycle with the stalling of DNA polymerase during replication due to the presence of such secondary structures. Endonucleases SLX1/SLX4 induce the formation of a DSB to remove the anomalous structures. The break will be solved during mitosis by the RAD51 recombinase, that binds 3′ end of the cleaved DNA strand and brings it near to the homologous template to complete DNA synthesis via POLD3/Polδ mechanism.

**Figure 3 ijms-23-05425-f003:**
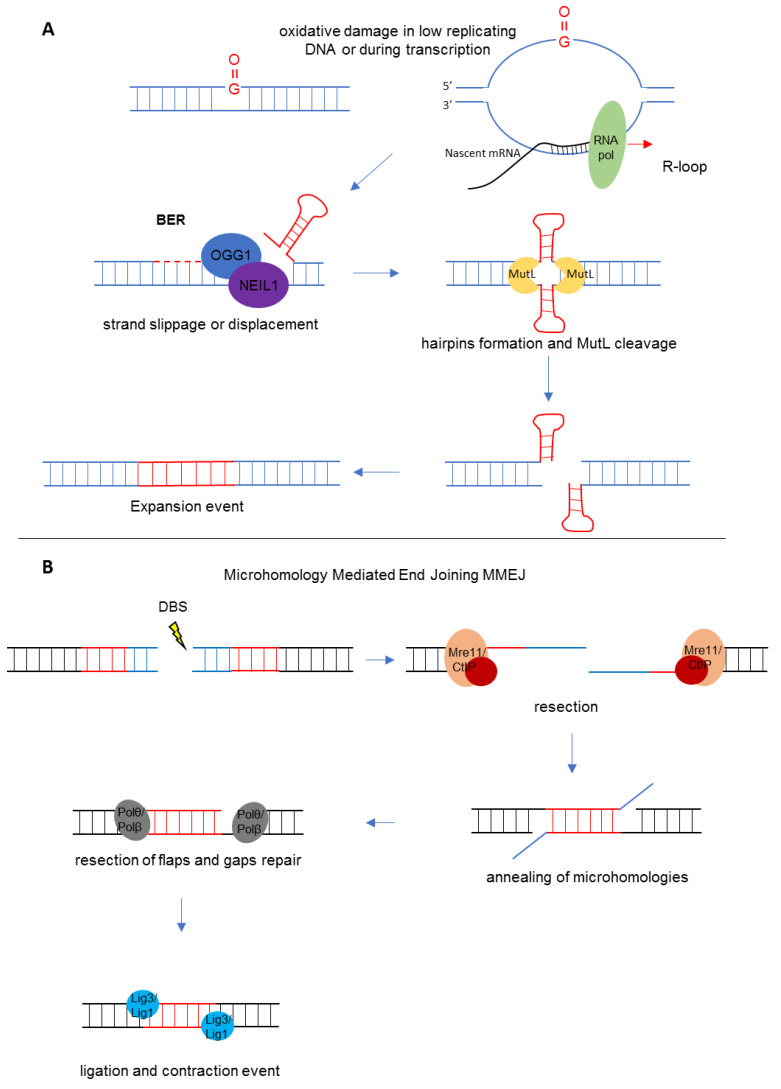
*P**roposed mechanisms of CGG instability in tissues with low rate of cell division.* (**A**) Oxidation of guanine in DNA strands or during transcription in the displaced DNA strand of the R loop activates BER mechanisms through the recruitment of OGG1 and NEIL1 enzymes, that remove the anomalous base. This damage generates a displacement of the DNA strand favoring hairpin formation. MutL enzymes bind the region containing hairpins in an attempt to remove them; the resulting cleavage produces a DSB which will be repaired, causing the repeat expansion. (**B**) Large contraction events could be addressed by MMEJ repair mechanism. A DSB may expose sequences of microhomology (MH, in blue) at both sides of the break, which may anneal through the removal of nonhomologous flaps at both ends. Polθ/Polβ complex will fill the gaps, which will be ligated by Lig3 and Lig1, resulting in contraction.

**Figure 4 ijms-23-05425-f004:**
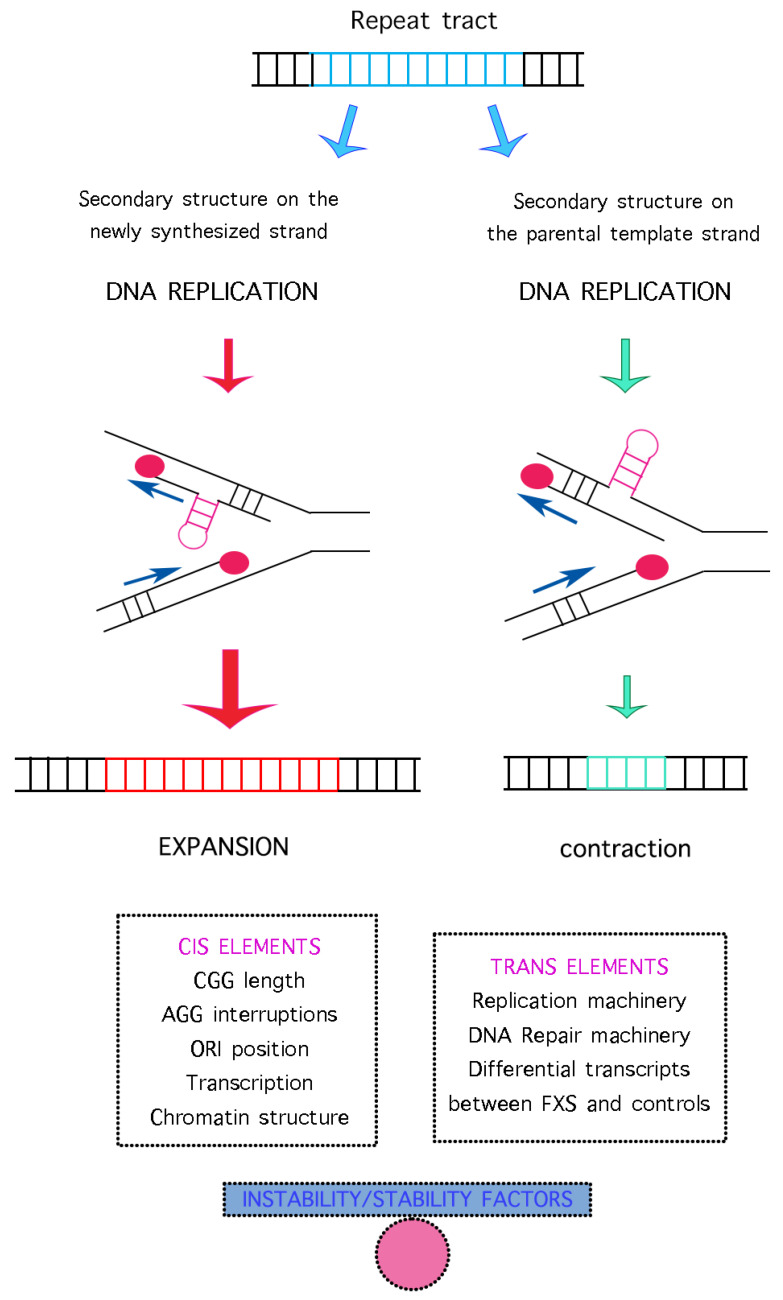
Factors that influence expansion and contraction events during DNA replication at the CGG repeat. The presence of the repeated CGG sequence (blue bars) on the nascent strand induces the formation of hairpins or other secondary structures. At the end of a second round of replication, an expansion (red bars) will result (on the left), whereas the presence of similar secondary structures on the template strand results in a stall with a late restart of the DNA polymerase that causes the exclusion of the repeated tract included in the secondary structure, thus leading to contraction events (green bars) after the second round of replication (on the right). The balance between listed cis and trans elements will hold the repeat sequence stable. An imbalance of these factors will favor instability (contractions or expansions).

## Data Availability

Not applicable.

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
