# Peer review of "Mechanisms of the FMR1 Repeat Instability: How Does the CGG Sequence Expand?"

_ijms, 2022, doi:10.3390/ijms23105425_

Round 1

Reviewer 1 Report

Tabolacci et al present a thorough presentation on the role of CGG/GCC expansion in FXR1. Overall architecture of the manuscript is solid. I warrant a few changes before its publication.

1) I would suggest the use of quadruplex (instead of tetraplex) just because the former is more common in the field.

2) When the authors depict the model of CGG G4s and GCC IMs, they present stacking of 4 units- is this scientifically correct? Is there evidence that both CGG and GCC structures make such stacks? or they form garlands? This needs to be CLARIFIED.

3) I would think having Figure 2 and relevant discussion before Figure 1 and relevant discussion would make more sense.

4) Citation inconsistencies, for example- (Gerhardt et al., 2014) in line 383.

Author Response

We answer point by point to the Reviewers requests.

Reviewer 1

1) We accordingly changed tetraplex with quadruplex throughout the text.

2) Although more structures than those reported may be actually formed by CGG·CCG sequence, the structures depicted in Figure 2 (formerly Figure 1) have been thoroughly investigated by Ajjugal et al., Sci Rep, 2021 quoted in text as ref. 37.

3) We agree with Reviewer 1 and quoted Figure 2 first. Previous Figure 1 is now Figure 2, while previous Figure 2 is now Figure 1.

4) The entire sentence that referred to Gerhardt et al., 2014 in line 383 has been deleted, as suggested by Reviewer 1.

We wish to thank Reviewer 1 for her/his precious work and for significantly improving our manuscript. We hope that in its present form it will be now suitable for publication in the International Journal of Molecular Sciences.

Reviewer 2 Report

The manuscript entitled "Mechanisms of the FMR1 repeat instability: How does the CGG sequence expand?" is presented a review of the various mechanism which plays a role in the expansion and contraction of FMR1 repeats. 

The manuscript is well organized, data is presented very well, no major issues identified.

 It will be good for the community if the testing algorithm for FMR1 is also included in the review. The authors are strongly encouraged to include it.

Author Response

We answer point by point to the Reviewers requests.

Reviewer 2

As suggested by Reviewer 2, we mentioned in the Introduction the testing algorithm for FMR1, including a new reference, number 6. Thus, previous references were accordingly renumbered in the text and in the references section.

We wish to thank Reviewer 2 for her/his precious work and for significantly improving our manuscript. We hope that in its present form it will be now suitable for publication in the International Journal of Molecular Sciences.